# Advancing Research: An Examination of Differences in Characteristics of Sexual and Non-Sexual Offense Recidivism Using a 10-Year Follow-Up

**DOI:** 10.3390/ijerph20136212

**Published:** 2023-06-24

**Authors:** Kristen M. Zgoba, Lin Liu, Dylan T. Matthews

**Affiliations:** Department of Criminology and Criminal Justice, Florida International University, Miami, FL 33199, USA; linliu@fiu.edu (L.L.); dmatthew@fiu.edu (D.T.M.)

**Keywords:** sexual offense recidivism, adverse childhood effects, mental health, substance use

## Abstract

In this article, we examine our current understanding of adverse childhood experiences (ACEs) and the intersection of mental health challenges and substance use on sexual and non-sexual recidivism. This study uses administrative data and comprehensive case files of a sample of 626 individuals who were incarcerated for a sexual offense. Each case was standardized to a 10-year post-release follow-up time for criminal history review. Findings reveal that ACEs, mental health challenges, and substance use disorder, along with a variety of other factors, affect the pathway to re-offending differently. Interestingly, despite the recent legislative push to utilize one standardized predictor of risk, individuals who commit sexual offenses (ICSOs) had very different re-offending patterns based on historic life events. This research will inform the current legislative debate by providing relevant empirical data on a large sample of ICSOs followed for a substantial period of time.

## 1. Introduction

Sexual offenses evoke strong reactions among communities and within popular culture. They represent the most disliked and unforgiven population of individuals within the criminal justice system, and that feeling extends to all types of individuals who have been labeled as “sexual offenders”, regardless of the circumstances and degree of the crime. Interestingly, unlike other types of individuals convicted of violent offenses, individuals convicted of sexual offenses (ICSOs) are often represented as singular, unique entities, different from all other crimes and criminals [1]. We see this view reflected in criminal law, where sexual crimes maintain distinct categories of criminal codes, but also through the post-sentencing sanctions that are levied against those found guilty of sexual offenses. Those with convictions for sex crimes are often required to participate in sex offender-focused treatment as a provision of their post-release supervision, and crime-specific penalties are imposed, such as lifetime registration and notification, residence restrictions, housing restrictions, and internet and travel bans [1]. Despite the fact that the empirical research demonstrates that ICSOs display a wide variety of behaviors and profiles, legislation, lawmakers and the public often collapse them into one type or group, furthering the presumption that those who commit sexual offenses all “look alike” and should all be dealt with in the same manner. 

What we know in reality is that ICSOs do not represent as a singular typology, nor do they necessarily specialize in sexual offenses [2,3,4]. While some individuals who commit sexual offenses maintain their victim choice across subsequent offenses, others tend to commit a variety of crimes and have a diversity of background characteristics [5,6,7]. As such, ICSOs do not always look and act the same. They have different backgrounds, demographics, and offending patterns. This heterogeneity has a range of implications for both research and practice [8]. This is important to note and is timely in light of many states’ consideration of the adoption of the Sex Offender Registration and Notification Act (SORNA; [9]), which standardizes the tiering of ICSOs across the country using a singular risk predictor, the crime of conviction [1,10,11].

In this article, we examine our current understanding of adverse childhood experiences (ACEs) and the intersection of mental health challenges and substance use on the discussion of sexual and non-sexual recidivism. This study will inform current legislative debate by using an extensive dataset spanning numerous decades of collection and providing relevant empirical data that identifies whether any characteristics cease in their importance, maintain significance after twenty-five years of research, or emerge as significant. Additionally, we will examine the index offense variable, considering its substantial influence on current sex offense registration and notification law. In sum, understanding which offenders are likely to sexually or non-sexually re-offend is complicated by the heterogeneity of offender characteristics, targets, characteristics, and nature of the crimes.

### Background

Although ICSOs generally fall into one legislative category in the United States, it is important to note that sexual offending can include a variety of criminal behaviors involving violence and nonviolence, adult or child victims (i.e., familial, known, or unknown/strangers), and physical contact (i.e., nonconsensual touch, rape) or no contact (i.e., exhibitionism, voyeurism, viewing child sexual exploitative material) [12,13]. The primary focus of sexual recidivism research efforts has been to identify (a) risk factors for re-offending [14,15,16] and (b) which offender characteristics are associated with sexual versus non-sexual re-offending [13,17]. Earlier meta-analyses and reviews of sexual recidivism have outlined the importance of a number of static and dynamic individual risk factors [18,19], most notably factors related to ICSOs’ deviant sexual interests and general criminal lifestyle/mentality [14,15,20,21,22]. Other factors influencing the rate of sexual recidivism may include the individual’s age, the type of sexual offense committed, previous offenses, the presence of underlying psychological challenges or substance use, previous trauma, and the level of social support [23]. 

Research shows that the risk for ICSOs to sexually recidivate is generally low relative to the length of the follow-up period (generally between 5% and 20%, even after 10 years of follow-up [22,24,25]). Moreover, the likelihood for ICSOs to recidivate with a non-sexual offense is significantly less compared with those who commit non-sexual offenses [13,17,26]. It is important to note that these findings can vary depending on the specific population being studied [12,27,28,29] and the length of time over which recidivism is measured [1]. For instance, most recently, out of a full sample of ICSOs under community supervision between 1986 and 2016 in Canada, ~32% were reconvicted for a new sexual crime, 44% for a new violent offense, and ~55% for any new offense after an extended follow-up period (mean = 20.5 years) [12]. For ICSOs deemed high risk/high need, on the other hand, about 40% were reconvicted for a new sexual crime, ~56% for a new violent crime, and about 66% for any new crimes. Despite what may be considered high rates of re-offending, the recidivism rates for these ICSOs are substantially lower over a longer period of time compared to individuals across 30 of the United States previously incarcerated in prison for any crime and released in 2005, approximately 68% were re-arrested within just three years, 79% within six years, and 83% within nine years [30].

Over the last few years, a growing body of empirical research has emerged pointing to the potential role adverse childhood experiences (ACEs) play in sexual offending [31,32,33,34,35,36]. Examples of ACEs include abuse (i.e., physical, emotional, sexual), neglect, abandonment, and other forms of family dysfunction (i.e., exposure to domestic violence or substance abuse). Individuals who are victims of traumatic events during childhood can have lasting persistent difficulties throughout life with their physical, mental, and emotional well-being. 

Early research on individuals who experienced ACEs has demonstrated these individuals had more psychological, behavioral, and health issues throughout their lives [37]. Having increased ACEs may be a risk factor for deviant or criminal behavior, including sexual offending [32,33,38,39]. For instance, individuals with more ACEs tend to be more likely to be serious, violent, and chronic offenders than those who experienced fewer ACEs [40,41]. In the case of sexual offenses, Levenson and Socia [26] surveyed ICSOs participating in civil abatement and outpatient treatment and found that those with more ACEs were also more likely to have an increased number and variety of previous arrests.

Other research demonstrates that ICSOs exhibit high instances of ACEs [33,42], more so than other convicted individuals [26,37,43,44]. For instance, DeLisi and colleagues [37] showed that confined male juveniles with at least two ACEs were more likely to have been charged with a sex offense (i.e., aggravated sexual assault, attempted aggravated sexual assault, attempted sexual assault, or sexual assault), and those with six ACEs were over five times more likely to have been charged with a sexual offense. In another study, Levenson et al. [44] demonstrated that male ICSOs were 13 times more likely to have been verbally abused, 4 times more likely to have endured emotional neglect, 3 times more likely to have been sexually abused, and about twice more likely to have been physically abused during their childhood compared with other incarcerated males within the general population. More recently, Craig and Zettler [31] showed that previously incarcerated juveniles with higher ACE scores were more likely to be re-arrested for a sexual assault than for other violent offenses. At minimum, having increased ACEs may negatively affect social functioning [35].

Additionally, mental health challenges, such as depression, anxiety, and personality disorders, can disrupt an individual’s ability to think rationally, control their impulses, and make healthy decisions, including decisions related to substance misuse. While it is true that not all people with mental health disorders or substance use disorders will commit violent or sexual offenses, extant research links these disorders with impaired judgment and engagement in risky behavior, including sexual re-offending. For example, in a study examining ICSOs released from prison, the odds to be reconvicted for a sexual offense after a follow-up period of between three and eight years were nearly double for ICSOs diagnosed with alcohol dependence, triple for those with drug use disorder, and ten times more for those with a personality disorder [45]. 

Recent research suggests that substance use disorders may play a more substantial role in predicting recidivism than mental illness [46,47,48]. Zgoba and her colleagues [48] showed that released individuals diagnosed with substance use disorder at the time of incarceration were at a higher risk of being re-arrested three years post-release than offenders without a substance use disorder, regardless of whether those who recidivated were diagnosed with a mental illness. These findings echo those of earlier work by Hanson and Bussière [22] who found no statistically significant differences in mental health-related factors between ICSOs who recidivated and those who did not.

Research also points to a possible connection between ACEs and issues related to mental health, substance use, and their co-occurrence with criminal offending [32]. In a sample of juveniles who completed community-based detention, Craig and colleagues [32] showed that both mental health and current drug use, as well as their comorbidity, partially mediated the positive relationship between ACE scores and re-arrest, suggesting some degree of interrelatedness. Similarly, Levenson [49] found that higher ACE scores were associated with increased substance misuse behavior in a sample of ICSOs seeking treatment. In addition, ICSOs with higher ACE scores were more likely to have used force, weapons, or physical violence during the commission of their offense [49]. 

In conclusion, the reviewed research on the influence of ACEs, mental health, substance use, and their co-occurrence suggests a nuanced relationship with sexual and non-sexual re-offending, in which causal or direct associations are still unclear. This article will build upon the existing gaps in ACE and co-occurring mental and substance disorders, along with other potential indicators, to determine their influence on re-offending patterns over a lengthy period of follow-up. Additionally, we explore whether there should be considerations beyond what is relied upon in recent legislative efforts to standardize an ICSO’s risk level [1].

## 2. Materials and Methods

### 2.1. Data Collection

Data were collected at a correctional facility devoted solely to individuals convicted of committing sexual offenses. Data collection occurred in 2018 and included administrative case-file reviews for males incarcerated at the facility from its inception in 1976 until 2010. The total study sample originally consisted of 777 individuals released from 1976 to 2010. This sample was later reduced to 626 cases due to a limited amount of missing data. Each individual had a 10-year standardized post-release follow-up time for criminal history review. Administrative case-record reviews were conducted by the first author, and a wide variety of data elements and indicators were extracted for each case in the study (approximately 450 data elements were collected for each case). The data included extensive demographic and descriptive background information (i.e., date of birth, education, marital status, race, substance use history, history of mental health challenges, service in the military, religious beliefs), index offense information (i.e., the relationship between the victim and the ICSO, type of offense, age and gender of the victim, the location of the offense, drug and alcohol use during the commission of the offense, whether the treatment provider reports the individual demonstrated remorse or denial), criminal history information (i.e., offense dates and types, number of prior arrests, convictions and incarcerations for sexual and nonsexual offenses), sentencing information (i.e., time served in days, disciplinary infractions, final charge disposition, release date), and childhood experiences (history of adverse childhood events, growing up in a single-parent home, familial involvement in the criminal justice system). 

The criminal history data were drawn from the State Police Computerized Criminal History System (CCH) and the National Crime Information Center’s Interstate Identification Unit (III). Through these sources, the researchers had the ability to track criminal history information for crimes committed as state, local, and municipal crimes, as well as for offenses that took place in all other U.S. jurisdictions. 

### 2.2. Measurements

#### 2.2.1. Dependent Variables

*Sex offense recidivism.* This dependent variable was coded as a binary variable (0 = no, 1 = yes) indicating whether the individuals in the sample were re-arrested due to another sex offense before the end of the 10-year observation window (ranging from ten years from the release date). As data collection and criminal histories were reviewed here, sexual offenses were defined by the United States New Jersey Code of Criminal Justice, Title 2C, 14. These offenses include all relationships and a wide variety of sexual acts. In New Jersey, a re-arrest generally results from a new offense rather than from a technical violation of community supervision/parole and is considered the individuals’ first stage of interaction with the criminal justice system. As such, re-arrest for a sexual offense is utilized as the measure for sexual recidivism in this study. For any offenses that took place outside the state of New Jersey, the crimes were coded from their classification and description.

*Non-sex offense recidivism*. This dependent variable was also measured as a binary variable (0 = no, 1 = yes), indicating whether the individuals in the sample were re-arrested due to a non-sex offense before the end of the 10-year observation window (ranging from ten years from the release date). Similar to the above, these offenses were coded as non-sexual by using the New Jersey Code of Criminal Justice, Title 2C. These offenses include all offenses that are not Title 2C, 14 offenses in the state of New Jersey. For any offenses that took place outside the state of New Jersey, the crimes were coded from their classification and description. As with sexual recidivism, a re-arrest generally results from a new offense rather than from fa technical violation of community supervision/parole and is considered the individuals’ first stage of interaction with the criminal justice system. As such, re-arrest for a non-sexual offense is utilized as the measure for non-sexual recidivism in this study. 

#### 2.2.2. Independent Variables

*Adverse Childhood Experiences (ACEs).* Following past practice [50], we created an additive scale of adverse childhood experiences based on these binary variables in the dataset: (1) growing up in a single-parent household; (2) growing up with family members who were involved in the criminal justice system; (3) reported experiences of sexual abuse during childhood; (4) reported experiences of physical abuse during childhood; (5) reported experiences of emotional abuse during childhood; (6) and experiencing other forms of abuse during childhood. This scale theoretically ranges from zero to six, depending on how many of the above experiences are present in the detailed administrative case files; however, no one in the sample had all six. The maximum value was four. A higher value of ACEs indicated that the individual experienced more types of adverse experiences during childhood.

*Co-occurring Disorders (CODs).* As a central thesis of this analysis involved the influence of CODs on recidivism, the researchers created four subgroups: (1) those with both mental health and substance use disorders (a COD); (2) those with mental health issues only; (3) those with substance misuse only; and (4) those with neither mental health nor substance misuse issues. The required information to create the COD designations was retrieved from the detailed information available to the researchers in the individual’s administrative case files. These files included therapy reports on mental health challenges and diagnoses, as well as reports of past and current substance use.

*Denial and Minimization*. This binary variable was created based on the individuals’ attitudes toward the sex offense that resulted in their current incarceration. The variable had a value of one when respondents denied themselves to be the aggressor and/or responsible for the crime and a value of zero otherwise. Similar to the other variables, the required information to create the dichotomy was retrieved from the detailed information available to the researchers in the individual’s administrative case files.

*Remorse*. Remorse was represented by a binary variable tapping into individuals’ expression of remorse for the sex offense that resulted in their current incarceration. The variable had a value of one when respondents felt remorse for the victim and a value of zero otherwise. Similar to the other variables, the required information to create the dichotomy was retrieved from the detailed information available to the researchers in the individual’s therapy case files. Often, this variable represented the opposite of the denial/minimization variable above. 

#### 2.2.3. Control Variables

To obtain an unbiased estimation of CODs and ACEs on recidivism, we controlled for the effect of potential confounding information. First, the research team controlled for the effect of military experiences. *Military experience* was a binary variable indicating whether individuals had military experiences on their records. The effect of parole supervision on recidivism was adjusted as well. *Parole supervision* was a binary variable indicating whether respondents were released under parole supervision. Furthermore, the number of disciplinary *prison violations/infractions* during the index sexual offense incarceration was a numeric variable indicating the count of prison violations. The *length of incarceration* was represented by the number of days the individuals were incarcerated during the index sexual offense. *Using a weapon* against the victim was a binary variable representing whether the ICSOs used a weapon against the victim during the index sex offense. This information was retrieved from the police report, and when a weapon was utilized, it was most frequently represented by a blade or firearm. Individuals’ criminal histories were measured by three variables: prior arrests (numeric count variable), having been arrested during adolescence due to a non-sex offense (binary variable), and having been arrested during adolescence due to a sex offense (binary variable). Intellectual disability was represented by a binary variable indicating whether respondents were diagnosed to have an intellectual disability in either their childhood or at the institutional intake assessment. Lastly, the researchers controlled for demographic factors, which included age (numeric count variable), race (1 = White, 2 = Black, 3 = Latino), marital status before the current incarceration (1 = married, 0 = not married), employment status before the current incarceration (1 = employed, 0 = not employed), parental status (1 = having children, 0 = having no children), and educational attainment (1 = equivalent or above a high school diploma, 0 = not graduated from high school). 

#### 2.2.4. Analytic Strategy

The analyses were conducted in three steps using Statistical Analysis Software (SAS) Version 9.3 (SAS Institute Inc., Cary, NC, USA). First, we examined the descriptive statistics of the sample to understand the distribution of the data. Second, we conducted a series of binary analyses to ascertain the effects of predictors on the acceleration of recidivism, represented by the length of time respondents “survived” from release to a re-arrest due to a new sex offense, without controlling for covariates. Lastly, using multivariate logistic regression, we examined the CODs’ and ACEs’ effects on both sexual and non-sexual offense recidivism, net of the effects of confounders. 

## 3. Results

### 3.1. Descriptive Statistics of the Sample

Table 1 reports the descriptive statistics of the sample. The largest proportion of participants was White (77%), followed by Black (21%) and Latino (2%). Approximately half of the sample (48%) was released under parole supervision and 35% of respondents did not finish high school. Twenty-three percent of individuals were diagnosed to have an intellectual disability. Approximately 18% of respondents had co-occurring mental health and substance use disorders, 21% experienced mental health disorders only, 25% suffered substance disorders only, and 36% suffered neither risk factors. ACEs were prevalent among individuals; more than half (57%) of the respondents experienced ACEs. On average, based on the ACE score created using 6 ACE items, respondents had an ACE score of 0.84 (min = 0; max = 4). After the current incarceration and over a 10-year follow-up period, approximately 23% of individuals committed another sex offense, and 30% committed a non-sex offense. 

### 3.2. The Acceleration of Recidivism by Predictors

To examine whether the predictors in this study were associated with divergent survival times and the time between release and the re-arrest, we employed survival analysis. After creating the Kaplan–Meier (KM) survival plots between each predictor and sex offense recidivism, we found that subgroups of respondents characterized by four predictors had distinct rates of re-arrests (Figure 1). These distinct groups included intellectual disability, education attainment, sexual abuse experiences during childhood, and emotional abuse experiences during childhood. The group with intellectual disability, represented by the red curve, fared better than their peers without intellectual disability, represented by the blue curve. The blue curve has a sharper drop, indicating that respondents in this group experienced re-arrests more quickly after release. Similarly, those with an educational attainment equivalent to or above a high school diploma experienced a faster re-arrest rate than their peers who did not graduate from high school. We also found that abuse experiences during childhood were associated with divergent rates of re-arrest. Respondents who did not experience sexual abuse demonstrated more accelerated re-arrests than those who experienced sexual abuse during childhood. Similarly, experiencing emotional abuse was associated with a slower pace of re-arrest. Other predictors were not found to be clearly associated with divergent re-arrest rates. While informative, these survival analyses were descriptive in nature. They tell us little about the effects of variables of primary interests on recidivism when a wide range of confounders are not taken into consideration. Therefore, we employed logistic regression to assess the association between ACEs and CODs on recidivism.

### 3.3. Logistic Regression Predicting Sexual and Non-Sexual Recidivism

We performed two logistic regression analyses to assess the predictors’ effects on sex offense and non-sex offense recidivism, respectively. Model 1 (Table 2) assessed the effects of predictors on sex offense recidivism. It is important to note that the index crime of conviction, the crime type (rape, child molestation, incest, etc.), was removed from all models due to its insignificance. However, multiple variables were found to be significantly associated with sex offense recidivism. First, CODs were significant predictors of sex offense recidivism. Compared with respondents who suffered from both mental health and substance use disorder, all three other subgroups had lower odds of sex offense recidivism. Among the pairwise comparisons between those with CODs and the other three subgroups, two pairwise comparisons achieved statistical significance. Respondents with neither mental health nor substance use disorders had significantly lower odds of sex offense recidivism compared with their peers with CODs (*OR* = 0.41, *p* < 0.01). Furthermore, although the difference was at a smaller scale, respondents with substance use disorders also only had significantly lower odds of sexual recidivism than those with CODs (*OR* = 0.55, *p* < *0*.05). Third, using a weapon against the victim was a significant predictor of a future sex offense. However, the direction of the effect was counterintuitive. Compared with those who did not use a weapon in their index sex offense, those who used a weapon had lower odds of recidivism (*OR* = 0.40, *p* < 0.05). Education achieved significance in the model as well. Those who did not graduate from high school were less likely to recidivate compared with peers who had an educational attainment equivalent to or above a high school diploma (*OR* = 0.64, *p* < 0.05). Lastly, we also found modest statistical evidence of two predictors’ effects on sex offense recidivism. There was modest evidence that those who committed sex offenses during adolescence were more likely to commit another sex offense after their release from current incarceration. Compared with those who did not have a record of sex offense arrest during adolescence, those who had this record were found to have higher odds of sex offense recidivism (*OR* = 1.61, *p* < 0.1). Similarly, the length of their current incarceration achieved significance in the model, but with modest statistical evidence. The length of incarceration was found to be slightly negatively associated with the odds of sex offense recidivism (*OR* = 0.998, *p* < 0.1).

Model 2 (Table 2) summarizes the effects of predictors for non-sex offense recidivism. Because the outcome for this model was non-sex offense recidivism, having a juvenile arrest record for a sex offense was replaced with having a juvenile arrest record for a non-sex offense. Additionally, we excluded the situational factor of using a weapon against the victim in the index sex offense. Several predictors achieved significance in the model. First, ACEs were significant predictors of non-sex offense recidivism. The higher the ACE score, the higher the likelihood of committing a non-sex offense after release. Each one-unit increase in the ACE score was associated with a 34% increase in the odds of committing a non-sex offense after release (*OR* = 1.34, *p* < 0.01). Furthermore, not all family factors functioned as suppressors of non-sex offense recidivism: having children protected, while married but no children, facilitated recidivism. For those who were married but had no children, their odds of recidivism were significantly higher than their counterparts who were single (*OR* = 1.73, *p* < 0.05). In contrast, having children was negatively associated with recidivism. Those who did not have children were found to have significantly higher odds of recidivism (*OR* = 2.46, *p* < 0.01). We also found that the number of prison violations/disciplinary infractions during the current incarceration predicted the risk of recidivism. Each one-count increase in the number of prison violations was associated with a 5% increase in the odds of recidivism for a non-sex offense (*OR* = 1.05, *p* < 0.05). Prior arrests were also associated with non-sex recidivism, with a one-count increase in prior arrests associated with a 5% increase in the odds of recidivism (*OR* = 1.05, *p* < 0.05). Admitting one’s guilt of the index sex offense was associated with a higher risk of non-sex recidivism, although the statistical evidence was modest. For those who admitted their culpability of the index sex offense, these were more likely to commit a non-sex offense after release (*OR* = 1.51, *p* < 0.1). Lastly, for demographic factors, race emerged as a significant predictor of non-sex recidivism. Compared with White respondents, Black respondents were found to have a 72% higher risk of recidivism for a non-sex offense (*OR* = 1.71, *p* < 0.05).

## 4. Discussion

Prior research demonstrates that individuals convicted of sexual offenses as a group do not necessarily fit one mold, despite legislation that assumes otherwise, nor do they similarly recidivate at comparable rates to other non-sexual offenders [13,17,26]. Additionally, research suggests that most ICSOs are not “specialists”. In other words, not only might they have a variety of prior arrests for other offenses, but they may go on to partake in a variety of future criminal behaviors that are not sexual in nature [12]. There are a number of associated risk factors that might lead an ICSO to commit a future offense, including the influential effects linked to mental health and substance use disorders, as well as any possible adverse childhood experiences (ACEs) [31,32,33,34,35,36]. The present analyses sought to determine which ICSO characteristics were more likely to be associated with individuals re-arrested for a sexual offense compared with those re-arrested for a non-sexual offense, especially characteristics related to ACEs, mental health, and substance use disorder. The results show that the risk factors associated with each type of recidivism are remarkably different and that ICSOs are more likely to commit a non-sexual re-offense over a 10-year follow-up period.

Descriptive findings revealed that overall recidivism rates for the sample of ICSOs were similar to those found in prior research, even after an average 10-year follow-up period (53% of individuals were re-arrested) [22,23,24,25]. For those who did recidivate, survival analyses showed four predictors that stood out in regard to the likelihood of general recidivism, albeit in ways not necessarily expected. For instance, ICSOs with at least one intellectual disability and those without at least a high school education took longer to be re-arrested than ICSOs that did not have an intellectual disability or those who completed high school. Additionally, ICSOs who experienced sexual or emotional abuse during childhood also took longer to be re-arrested than those who were not abused. Taken together, these findings illustrate that ICSOs who take longer to recidivate are more often individuals who otherwise should have more difficulties in assimilating back into society (e.g., co-occurring mental and substance use disorder, do not have a high school education, and had at least some adverse childhood experience). On the other hand, ICSOs most at-risk to recidivate sooner from the time of release do not look very different from the average community member (e.g., do not have an intellectual disability, have at least a high school education, and were not sexually or emotional abused as a child). If preventing recidivism continues to be a primary criminal justice initiative, it is important to match post-incarceration restrictions and support relative to the needs of the individual released, as opposed to treating all ICSOs exclusively based on their index offense.

Next, logistic regression models were utilized to determine which individual characteristics were associated with sexual recidivism and which were associated with non-sexual recidivism. The first model for sexual offense recidivism showed that ICSOs without either mental health or substance use disorder are least likely to be re-arrested for a sexual offense, whereas those experiencing co-occurring mental health and substance use disorder are most likely to recidivate. This finding supports the accepted risk assessment strategies, in which the concentration of supportive services ought to be matched to the criminogenic needs of each released individual rather than required of all ICSOs [18,19]. The central implication here is that, unless there is a pattern of sexual deviancy prior to the index offense, it may be more beneficial to address mental health- and substance use-related issues rather than singling ICSOs out for sexually deviant behaviors only, particularly because ICSOs go on to more frequently commit non-sexual recidivism.

The variables that resonate as significant for sexual recidivism over a 10-year follow-up period demonstrate as less criminogenic in nature, meaning that other than having a COD, the other variables are not considered “typical” predictors or their direction is somewhat counterintuitive. For example, ICSOs who were more likely to recommit a sexual offense demonstrated higher levels of education and did not use a weapon. Similarly, using a nationally representative sample of incarcerated men, one recent study showed that individuals with a high school diploma had a 1.5 times increased risk for a sexual conviction compared with individuals without one [51]. It may be that education is not a driving force but is operating more as a characteristic of the overall group of ICSOs. Interestingly, those who used weapons during the commission of the sexual crime also demonstrated lower sexual recidivism rates, a counterintuitive finding. One would expect those who brandish weapons are likely to be more violent and, hence, more recidivistic, but this was not the case. There is a possibility that because ICSOs who utilized weapons during the commission of the sexual crime received longer sentences (forty percent longer sentences), there may be a latent effect of time and age on recidivism. Most important to the discussion on using the offense type (rape, child molestation, incest, etc.) or crime of conviction as a main predictor of subsequent re-offending was its lack of significance in any bivariate or multivariate relationships with recidivism. This is concerning, given how recent legislation is turning toward standardizing risk prediction by using this singular indicator [11]. Standardizing its use and potentially affecting the registration status and timeframe of hundreds of thousands of ICSOs is likely a disservice to the safety and security of the public, thereby applying a “one size fits all” solution to a problem that is clearly multifaceted.

Although the association between ACEs and sexual recidivism was not statistically significant, the second model for non-sexual recidivism revealed several statistically significant associations. The model for non-sexual recidivism showed that ICSOs with higher ACE scores were more likely to be re-arrested for a non-sexual crime than those who scored lower. This finding demonstrates that ICSOs with increased ACEs are more likely to exhibit various criminal behaviors, thus representing a need to address the underlying criminogenic issues one may be dealing with rather than solely addressing behavior related to the index crime or deviant sexual behavior in general. These findings support those found regarding a more generalized context of criminal [37] rather than sexual recidivism specifically, in which individuals with more ACEs tend to be more chronic offenders than those with fewer or none [40,41].

Additionally, the results confirm more traditional risk factors associated with criminal recidivism relating to prior criminal behavior [14]. Specifically, the odds to be re-arrested for a new non-sexual offense are significantly increased for individuals with more prior arrests and more behavioral violations while incarcerated, as well as for those who admitted their guilt for the index offense, compared with ICSOs that were not re-arrested. Furthermore, these findings suggest that for re-arrested ICSOs, it may be the case that their likelihood to recidivate is the result of an overall pattern of criminal behavior rather than some degree of crime specialization [14,15,20,21,22,23]. The implication here is that grouping all ICSOs under one legislative category undermines the associated individual risk factors for the different types of recidivism. It may also be that the post-incarceration experiences for minority ICSOs differ from those compared with non-minority ICSOs. We found that Black ICSOs were more likely to be disproportionately re-arrested for a non-sexual offense compared to White ICSOs. This is supported by the vast majority of criminological research, which indicates that individuals holding a minority status maintain higher rates of interaction with law enforcement. This raises the question of whether these individuals commit more offenses or are disproportionately monitored and arrested. For the purposes of this study, it is important to examine the post-incarceration setting and available support for re-entering ICSOs. Future research would benefit from specifically analyzing post-incarceration and support settings.

A final implication raised from the findings is illustrated in the form of familial support post-incarceration. The results from the second model for non-sexual re-arrests suggest that familial support when received in the form of a spouse is not enough to prevent recidivism. Compared with ICSOs released who were single, those who were married without children were more likely to be re-arrested. On other hand, having to provide familial support to one’s children acted as a protective factor against recidivism for released ICSOs, in that ICSOs with children were less likely to be re-arrested for a non-sexual offense than those without. As stated previously, future studies may benefit from examining these dynamic post-incarceration factors.

Lastly, this study is not without limitations. First, conducting a study of this type with sensitive sexual arrest data introduces a number of challenges. The most noted problem plaguing sexual offense research, i.e., the low base rate of reported sexual offenses, is likely tied to the under-representation of official data. Because sexual offenses are likely under-reported, as are many crimes, most measures of recidivism will under-represent the true offending rates [22,27,52,53]. While this has been recognized in much recent research, it has also been noted that some types of sexual abuse may be over-represented to the police, such as stranger rapes [53,54,55]. This challenge is also likely linked to a second limitation regarding the use of official law enforcement data. Official sources of data are the most relied-upon sources of data for research that documents historic crime counts; however, not all crimes come to the attention of law enforcement. Self-reports of offending behavior and victimization surveys offer alternatives but are considered anecdotal and less definitive in their findings. As such, official records will continue to be the most heavily used, while noting their potential drawbacks. Lastly, the data used in this study are both historic and come from correctional institution placements. Therefore, findings from the present study might not generalize to lower-risk or non-incarcerated ICSOs and definitive correlates between the independent and dependent variables may be obscured.

## 5. Conclusions

Despite the fact that empirical research shows that ICSOs exhibit a wide range of behaviors and characteristics, lawmakers and the general public frequently aggregate them into one category or group, supporting the idea that sexual offenders should all be treated equally because they all “look alike.” If categorizing offenders into known risk classes is going to be the basis for decision-making, such classifications must be established by empirically proven processes that are likely to correctly identify higher risk offenders. This study demonstrated that the indicator being used to standardize risk through the Sex Offender Registration and Notification Act (SORNA; [9]) was not a significant predictor of either sexual or non-sexual recidivism. The fact that ICSOs demonstrated heterogeneity in their re-offense patterns has a range of implications for both research and practice. To determine which ICSOs pose the greatest risk to public safety, empirically derived risk assessment models based on variables known to correlate with recidivism should be employed. The evaluation of individual risks and needs, as well as ICSO support systems, and collaboration with treatment providers and law enforcement will aid in risk management. Public safety will be enhanced with the utilization of empirically supported results and collaboration among stakeholders, as opposed to dependence on excessively broad one-size-fits-all policies.

## Figures and Tables

**Figure 1 ijerph-20-06212-f001:**
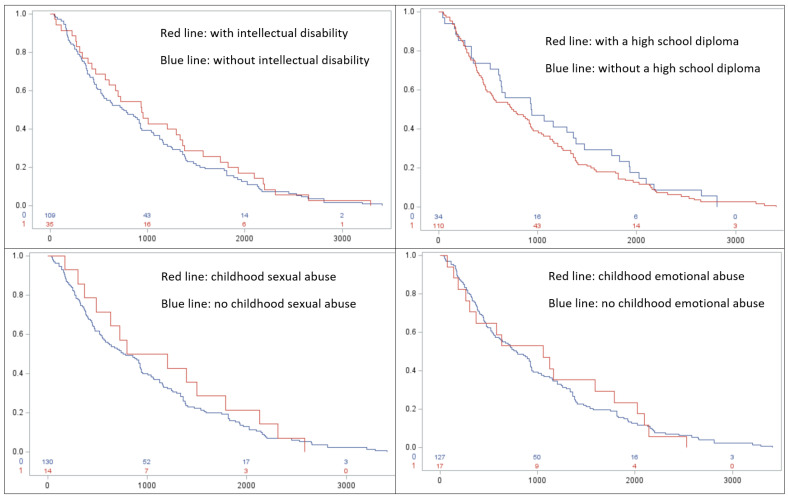
Survival curve of subgroups over a period of 3650 days after release.

**Table 1 ijerph-20-06212-t001:** Descriptive Statistics (*n* = 626).

Variables	Mean or Percentage	SD	Min	Max
Dependent variables				
Sex offense recidivism	23.20%			
Non-sex offense recidivism	30.15%			
Independent variables				
ACE score	0.84	0.91	0.00	4.00
COD subgroups				
No mental health or substance use issues	35.7%			
Mental health issues only	21.6%			
Substance use issues only	24.74%			
Co-occurring mental health and sub-stance use issues	18.3%			
Denial	29.61%			
Remorse	35.07%			
Juvenile non-sex offense history (yes)	21.39%			
Juvenile sex offense history (yes)	14.69%			
Control Variables				
Age at release	31.38	80.21	18	75.01
Hispanic	2.08%			
Black	20.94%			
White	76.98%			
Graduated from high school	64.95%			
Intellectual disability	23.2%			
Having children	47.84%			
Married	31.44%			
Prior arrests	2.67	4.13	0.00	54.00
Served in the military	48.45%			
Served on parole supervision	48.2%			
Number of prison violations	1.59	3.84	0.00	48.00
Incarceration length (days)	1526.23	1977.22	0.00	25,924.00
Used weapon during the sex offense	8.89%			
Employed before the current incarceration	85.05%			

**Table 2 ijerph-20-06212-t002:** Logistic Regression Results on the Effect of ACEs and CODs.

	Model 1: Sex Offense Recidivism		Model 2: Non-Sex Offense Recidivism	
	OR	95% C.I.	OR	95% C.I.
ACE score	1.18	(0.95–1.47)	1.34 **	(1.09–1.66)
No mental health or substance use issues	0.42 **	(0.24–0.73)	0.67	(0.36–1.25)
Mental health issues only	0.83	(0.47–1.47)	0.97	(0.54–1.76)
Substance use issues only	0.55	(0.30–1.00)	1.05	(0.54–2.01)
Denial	1.16	(0.74–1.82)	1.52 †	(0.97–2.38)
Remorse	1.32	(0.86–2.03)	1.24	(0.82–1.89)
Juvenile non-sex offense history	0.84	(0.51–1.37)	0.75	(0.47–1.21)
Juvenile sex offense history	1.61	(0.93–2.77)		
Age	1.00	(0.98–1.02)	0.99	(0.98–1.01)
Black	0.75	(0.44–1.26)	1.72 *	(1.06–2.77)
Hispanic	1.26	(0.31–5.19)	1.95	(0.49–7.44)
Served in the military	1.28	(0.84–1.94)	1.06	(0.70–1.59)
Graduated from high school	0.64 *	(0.41–1.00)	0.93	(0.61–1.41)
Intellectual disability	0.77	(0.47–1.25)	1.26	(0.76–2.09)
Having children	1.36	(0.84–2.20)	2.46 **	(1.52–3.99)
Married	1.08	(0.65–1.80)	1.75 *	(1.06–2.89)
Prior arrests	0.96	(0.91–1.02)	1.05 †	(1.00–1.11)
Number of prison violations	1.03	(0.98–1.07)	1.05 *	(1.00–1.11)
Incarceration length	1.00 †	(1.00–1.00)	1.00	(1.00–1.00)
Used weapon during the sex offense	0.40 *	(0.17–0.94)		
Employed before the current incarceration	1.32	(0.75–2.32)	1.06	(0.62–1.80)

The reference category for race/ethnicity is White, and the reference category for the COD group is the group with co-occurring mental health and substance misuse issues. † *p* < 0.1, * *p* < 0.05, ** *p* < 0.01.

## Data Availability

Not applicable.

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
