# Peer review of "Advancing Research: An Examination of Differences in Characteristics of Sexual and Non-Sexual Offense Recidivism Using a 10-Year Follow-Up"

_ijerph, 2023, doi:10.3390/ijerph20136212_

Round 1

Reviewer 1 Report

Minor revisions for authors:

Line 24 - sentence needs support; citation. May want to look at Stephen Holmes on typologies of pedophiles & hebephiles.

Line 80 - needs citation

Line 202 - Why use NJ code? Should insert further explanation and reasoning - this primarily for scholars from other countries who may not understand the US/State penal codes. 

ACE  - very good scale and additives completely related to research.

Chart and results illustrations are excellent and easily readable and understandable. 

Line 390 - recommend but not necessary to put quotes around "specialists". I would expand on the explanation for readers for a better understanding.

Author Response

We thank the reviewer for reading and commenting on our article. It has been improved due to these comments. Please find each comment below and our response to it. It is important to note that the reviewer and the authors show different line numbering (i.e., on my version, line 24 is not where I would think the Holmes citation would go since that sentence is not about typologies), so we have done our best to place the changes where we believe the reviewer wanted them. 

Line 24 - sentence needs support; citation. May want to look at Stephen Holmes on typologies of pedophiles & hebephiles.

COMMENT: Thank you, this was added around line 40 (on our version) where we mention typologies. This was added to the document text and the reference list. We added:  Holmes, R. M.; Holmes, S. T. Profiling violence crimes: An investigative tool (4th ed.) 2009, Sage: Thousand Oaks, CA. https://psycnet.apa.org/record/2008-13749-000. 

Line 80 - needs citation.

COMMENT: Thank you. Line 80 had citations on our version, but we looked around there and the only sentence that didn't have one is: "Other factors influencing the rate of sexual recidivism may include the individual’s age, the type of sexual offense committed, previous offenses, the presence of underlying psychological challenges or substance use, previous trauma, and the level of social support."

The below reference was added to the document text and the reference list. We added, Hanson, R. K., & Morton-Bourgon, K. (2004). Predictors of sexual recidivism: An updated meta-analysis 2004-02. Public Safety and Emergency Preparedness Canada.

Line 202 - Why use NJ code? Should insert further explanation and reasoning - this primarily for scholars from other countries who may not understand the US/State penal codes. 

COMMENT: Thank you for bringing this to our notice, we had not considered this and appreciate you noting it. Because the criminal history data were collected in New Jersey, United States, the NJ criminal codes and laws are discussed. I have added the appropriate context to the manuscript.  

ACE  - very good scale and additives completely related to research.

COMMENT: Thank you very much. 

Chart and results illustrations are excellent and easily readable and understandable. 

COMMENT: Thank you very much. 

Line 390 - recommend but not necessary to put quotes around "specialists". I would expand on the explanation for readers for a better understanding.

COMMENT: Thank you, we added the quotes and have the sentence below after the word specialists: ‘specialists’.  "In other words, not only might they have a variety of prior arrests for other offenses, but they may go on to partake in a variety of future criminal behaviors that are not sexual in nature [11]."

Reviewer 2 Report

Thank you for the opportunity to read the article. As a scholar of cultural studies, I would comment the contents only for those parts that fall into my field of expertise.

I would have only one comment regarding the contents.

"It may also be that the post-incarceration experiences for minority ICSOs differs from those compared to non-minority ICSOs. We found black ICSOs were more likely to be disproportionately rearrested for a non-sexual offense compared to white ICSOs."

Maybe here should be asked the question are black people in general more easily arrested or what are the reasons why they would be more frequently rearrested.

Author Response

Thank you for your review of the article, "Advancing Research: An Examination of Differences of Correlates of Sexual and Non-Sexual Offense Recidivism Using a 10-Year Follow-up". The reviewer lists that he/she only has one comment:  

"It may also be that the post-incarceration experiences for minority ICSOs differs from those compared to non-minority ICSOs. We found black ICSOs were more likely to be disproportionately rearrested for a non-sexual offense compared to white ICSOs."

Maybe here should be asked the question are black people in general more easily arrested or what are the reasons why they would be more frequently rearrested.

Comment from the authors: Thank you, we have added the following information to this point on page 11 of 15:

"It may also be that the post-incarceration experiences for minority ICSOs differs from those compared to non-minority ICSOs. We found black ICSOs were more likely to be disproportionately rearrested for a non-sexual offense compared to white ICSOs. This is supported by the vast majority of criminological research, which indicates individuals holding minority status maintain higher rates of interaction with law enforcement. This begs the question of whether these individuals commit more offenses or are disproportionately monitored and arrested. For purposes of this study, it is important to examine the post-incarceration setting and available support for reentering ICSOs. Future research would benefit from specifically analyzing post-incarceration and support settings."